# Dynamics of Treatment Response to Faricimab for Diabetic Macular Edema

**DOI:** 10.3390/bioengineering11100964

**Published:** 2024-09-26

**Authors:** Katrin Fasler, Daniel R. Muth, Mariano Cozzi, Anders Kvanta, Magdalena Rejdak, Frank Blaser, Sandrine A. Zweifel

**Affiliations:** 1Department of Ophthalmology, University Hospital Zurich, University of Zurich, 8091 Zurich, Switzerland; 2Division of Eye and Vision, Department of Clinical Neuroscience, Karolinska Institutet, 171 77 Stockholm, Sweden; 3St. Erik Eye Hospital, 171 64 Solna, Sweden; 4Eye Clinic, Department of Biomedical and Clinical Sciences, Luigi Sacco Hospital, University of Milan, 20157 Milan, Italy

**Keywords:** faricimab, anti-VEGF, ang-2, diabetic macular edema, DME, intraocular inflammation, IOI, intravitreal therapy, IVT, IVI

## Abstract

This study analyzes the dynamics of short-term treatment response to the first intravitreal faricimab injection in eyes with diabetic macular edema (DME). This retrospective, single-center, clinical trial was conducted at the Department of Ophthalmology, University Hospital Zurich. Patients with treatment-naïve and pretreated DME were included. Patient chart data and imaging were analyzed. Safety and efficacy (corrected visual acuity (CVA), central subfield thickness (CST), and signs of intraocular inflammation (IOI)) of the first faricimab intravitreal therapy (IVT) were evaluated weekly until 4 weeks after injection. Forty-three eyes (81% pretreated) of 31 patients were included. Four weeks after the first faricimab IVT, CVA remained stable and median CST (µm) decreased significantly (*p* < 0.001) from 325.0 (293.5–399.0) at baseline to 304.0 (286.5–358.0). CVA at week 4 was only associated with baseline CVA (*p* < 0.001). CST was the only predictive variable (*p* = 0.002) between baseline and week 4 CST. Weekly safety assessments did not show any sign of clinically significant IOI. This study suggests faricimab is an effective treatment for (pretreated) DME, showing structural benefit 1 month following the first injection without short-term safety signals.

## 1. Introduction

Intravitreal anti-vascular endothelial growth factor (VEGF) treatment is the standard of care for patients with center-involving diabetic macular edema (DME) [1]. With the launch of different anti-VEGF drugs (i.e., ranibizumab, aflibercept 2 mg, brolucizumab, faricimab), biosimilars (i.e., biosimilars to ranibizumab), and higher dosage protocols (i.e., aflibercept 8 mg), therapeutic options continue to grow. When selecting an effective treatment for a specific clinical scenario, it is important to consider a new drug’s efficacy relative to existing options, cost-effectiveness, health insurance coverage, and possible (novel) side effects [2,3,4,5]. Specifically, intraocular inflammation (IOI) is a relevant side effect with newer anti-VEGF drugs—potentially caused by pre-existing and treatment-emergent anti-drug antibodies along with the molecular structure of the drug [6,7]. Faricimab, the first bispecific intraocular antibody, targeting VEGF-A and angiopoetin-2 (Ang-2), received FDA approval in January 2022 after demonstrating non-inferior structural and functional results compared to aflibercept 2 mg in two phase 3 studies (YOSEMITE and RHINE) [8]. Intraocular inflammation incidence was low (2%) with no cases of retinal vasculitis—the most severe and sight-threatening form of IOI [6,8]. Post hoc meta-analysis and real-world data (RWD) confirm superior efficacy with comparable safety outcomes at 12 months [9,10]. However, there is emerging evidence of faricimab-induced (occlusive) vasculitis, something recognized by Genentech in a post-marketing safety warning in November 2023 [11,12,13]. Currently, little data exist about faricimab’s short-term efficacy or the timeline of IOI incidence after the beginning of treatment. This study investigates the dynamics of short-term treatment response one month following the first faricimab injection in patients affected by center-involving DME with weekly assessments for a subgroup of eyes.

## 2. Materials and Methods

### 2.1. Ethics

Ethics Committee approval was obtained from the local Ethics Committee of the Canton of Zurich, Switzerland (project no. PB_2016_00264). This study adheres to the tenets of the 1964 Declaration of Helsinki and its later amendments.

### 2.2. Study Design

This is a retrospective, single-center, clinical trial conducted at the Department of Ophthalmology, University Hospital Zurich, University of Zurich, Zurich, Switzerland.

### 2.3. Data Collection

Clinical patient data were extracted from the electronic patient chart system (KISIM, CISTEC AG, Zurich, Switzerland) and the imaging viewers Heidelberg Eye Explorer (Heidelberg Engineering, Heidelberg, Germany), Nikon Optos Viewer (Optos Inc., Marlborough, MA, USA), Solix Viewer (Visionix International SAS, Pont-de-l’Arche, France), and Zeiss Plex Elite Viewer (Carl Zeiss Meditec AG, Jena, Germany). Patients with DME were enrolled. For eyes previously treated with intravitreal anti-VEGF drugs (i.e., ranibizumab, aflibercept, bevacizumab) the criterion for switching was a treatment interval ≤ 6 weeks (“high-demanders”). Patients with previous intravitreal brolucizumab were excluded due to its reported incidence of IOI which could confound the objective of this study. Prior steroid (intravitreal, para-/retrobulbar) therapy, macular laser treatment, photodynamic therapy (PDT), or pars-plana vitrectomy eyes were also excluded from the study. The first faricimab injection date was defined as “baseline”. For a subset of patients, follow-up examinations were performed weekly until the second faricimab injection was administered after 4–5 weeks. Corrected-visual acuity (CVA) with auto-refraction (Nidek NT-530/510, Nidek Company, Ltd., Hirioshi-cho, Gamagori, Aichi, Japan) or current glasses was tested according to the Early Treatment of Diabetic Retinopathy Study (ETDRS) chart. Intraocular pressure (IOP) was measured by air puff tonometry (Nidek Company, Ltd., Hirioshi-cho, Japan) or Goldman applanation tonometry (Haag-Streit Group, Köniz, Switzerland). Further clinical data on drug safety were extracted from the patient charts; this included anterior chamber (AC) cells within a 1 mm × 1 mm slit beam field and AC flare as defined by the Standardization of Uveitis Nomenclature (SUN) Working Group [14]. Vitreous cells were assessed clinically in mydriasis at the slit lamp within a 1 mm × 1 mm slit beam field using a 78D, 66D, or Digital Widefield lens (Volk Optical, Mentor, OH, USA). The retinal vessel status was assessed clinically based on vessel perfusion, vessel caliper, tortuosity, and bleedings using the same fundoscopy lenses.

### 2.4. Image Analysis

Retinal scans were performed by spectral-domain optical coherence tomography (SD-OCT) (Spectralis, Heidelberg Engineering, Heidelberg, Germany). Fundus imaging was carried out with ultra-widefield color scanning laser ophthalmoscop (SLO) (California, Optos Inc., Marlborough, MA, USA). Image quality, ETDRS grid foveal centration, and segmentation boundaries were reviewed on all structural OCT scans. Manual corrections were performed, if necessary. As a structural-anatomical correlate for treatment efficacy, the thickness (CST) in the central subfield of the central 1 millimeter circle of the ETDRS grid was measured in micrometers [µm].

### 2.5. Statistical Analysis

The normal distribution of the quantitative variables was tested before the analysis using the Shapiro–Wilk test and Q-Q plots.

Descriptive statistics were summarized as median (IQR, 25th–75th interquartile range) or frequency number (%), as applicable. For normally distributed variables, mean ± standard deviation (SD) was presented.

For discrete variables, a paired Wilcoxon signed-rank test (not-normally distributed variables) or a paired samples *t*-test (normally distributed variables) were used to compare VA, CST, and IOP from baseline to week 4 within the study group.

The Kruskal–Wallis non-parametric test was used to compare the medians of the weekly assessments. A Chi-square (χ^2^) test was calculated within the Kruskal-Wallis test for the present degrees of freedom (df). For parametric distribution, the analysis of variance (ANOVA) was adopted.

Linear mixed-effects models were used to test predictor variables (VA and CST at baseline, number of previous treatments, and treatment naïve subgroup) against VA and CST at the end of follow up.

All statistical analyses were performed using R software (version 4.1.1, R Foundation for Statistical Computing, Vienna, Austria, https://www.R-project.org/, last accessed on 16 September 2024). A *p*-value < 0.05 was considered statistically significant.

## 3. Results

### 3.1. Demographics

The study included 43 eyes from 31 patients (20 males). The median age was 66 years (58.5–71.5). Out of the 43 eyes, 8 (18.6%) were treatment-naïve while the remaining eyes had received a median of 16 aflibercept or ranibizumab injections (6–28) prior to switching to faricimab. A subgroup of 23 eyes was assessed weekly after the first faricimab injection through to week four to collect efficacy and safety data. Table 1 presents a summary of the study sample demographics and ocular characteristics at baseline.

### 3.2. Efficacy

After one month, CVA remained stable compared to baseline, with the baseline median at 72 (68.5–78.5) versus the one month median at 75 (70–79) letters (Wilcoxon signed-rank test, *p* = 0.1). The median CVA was not significantly different between the four different time points of the weekly follow-ups, χ^2^(4) = 1.2 (Kruskal–Wallis test, *p* = 0.9). Figure 1A presents the mean difference in CVA (ETDRS letters) at various time points.

In the linear mixed-effects model, baseline CVA was the only predictor variable significantly associated with CVA at week 4 (*p* < 0.001). The beta coefficients and 95% confidence intervals of all variables are displayed in Figure 2A.

The central subfield thickness (CST) showed a statistically significant decrease after one month compared to baseline, with a median of 325.0 [IQR 293.5–399.0] vs. 304.0 [IQR 286.5–358.0] microns, (Wilcoxon signed-rank test, *p* < 0.001) (Figure 1B). Testing of only switched eyes (n = 35) confirms the decreased CST after one month (Wilcoxon signed-rank test, *p* < 0.01). For the subgroup of patients with weekly assessments, the results were not statistically significant χ^2^(4) = 4.15, (Kruskal–Wallis test, *p* = 0.39).

To address inter-subject correlation, the tests were repeated including only one randomly selected eye per patient (eyes n = 31, patients n = 31) with unchanged results: (Wilcoxon signed-rank test, *p* = 0.3 for VA, *p* = 0.02 for CST).

In the linear mixed-effects model, baseline CST was the only variable significantly associated with CST at week 4 (*p* = 0.002). The number of previous injections and patient treatment history (treatment-naïve or not) did not affect the outcome variable. Figure 2B presents beta coefficients and 95% confidence intervals of all tested variables.

### 3.3. Safety

The safety profile of faricimab in the study cohort was monitored weekly for a subgroup of 23 eyes. At week 2 and 3, there were three eyes with missing data due to missed appointments. Two eyes (8.7%) had 0.5+ AC cells according to the SUN scheme at week one, one eye out of 20 (5%) at week two, and two eyes out of 20 (10%) at week three. At week four, 0.5+ AC cells were noted in one eye (2.3%). No flare, vitreous cells, or vasculitis were detected at any assessment point in any eye. IOP remained stable throughout the follow-up period, with no significant change observed between baseline and week 4 (paired samples *t*-test, *p* = 0.11) (Figure 3). Sensitivity analysis with paired samples *t*-test, including only one randomly selected eye per patient, did not change the result (*p* = 0.1).

## 4. Discussion

Our study demonstrates a structural response to faricimab IVT in a cohort primarily consisting of eyes pretreated with short intervals due to recurrent or persistent DME, with no new safety signals observed within the first month.

Visual acuity remained stable after 4 weeks, which is to be expected in a cohort of mainly pretreated eyes (77%, median of 16 previous injections) [10,15]. However, CST significantly decreased with a median of 11 μm within the first 4 weeks, confirming the other RWD of improved structural outcomes in non-treatment naïve eyes even after one injection [15]. Improvement after switching from other anti-VEGF agents has been shown before, but not yet in a high demand cohort after the first injection [15,16]. Possible explanations for the structural improvement under faricimab include its dual blockade pathway of VEGF-A/Angiopoietin-2 and tachyphylaxis/tolerance to previously used drugs (rendering them less effective) [15,16]. Another explanation, or rather a bias in our data, would be the shortened interval after switching. However, as the median prior treatment interval of our cohort was 4 weeks, we do not believe we induced a significant treatment interval bias in our study. There remain points to debate considering faricimab for DME under real-world conditions; notably, whether it is advisable to follow a strict 4-weekly loading phase after switching from previous anti-VEGF and how much long-term treatment burden can be alleviated with extended treatment intervals—specifically, for pretreated eyes with chronic DME. These questions can only be answered with time, as more long-term RWD become available. The limitations to our study are its relatively small sample size and retrospective design.

The safety profile during the 4 weeks after the initial faricimab treatment was good with no signs of relevant IOI (only minimal anterior chamber cells on dilated exam and stable IOP). While this is consistent with an analogous study for AMD patients, to our knowledge this is the first study that assessed the safety of faricimab in DME on a weekly basis [17]. However, the follow-up time is too short to draw definitive conclusions about immunogenicity. Previous reports of severe IOIs have shown its occurrence within the first 6 months of treatment and not necessarily after the first administration [11,13]. As mechanisms of immunogenicity in anti-VEGF drugs are still not perfectly understood (possibly due to anti-drug antibodies, protein source, molecular drug structure, drug impurities, injection material impurities, procedure-related factors, patient factors, and/or others), and might be different for different types for IOI (acute inflammation/sterile endophthalmitis versus delayed onset inflammation/vasculitis), further RWD with longer follow-up should shed more light on the mechanisms and incidence of IOIs due to anti-VEGF drugs [6,7].

## Figures and Tables

**Figure 1 bioengineering-11-00964-f001:**
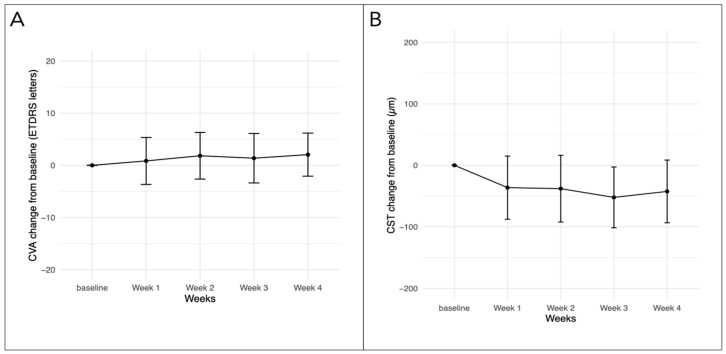
Change in CVA (**A**) and CST (**B**) from baseline for eyes with weekly follow-up (baseline, week 1 and week 4: n = 23, week 2 and week 3: n = 20). CVA—corrected visual acuity, CST—central subfield thickness, ETDRS—early treatment diabetic retinopathy study.

**Figure 2 bioengineering-11-00964-f002:**
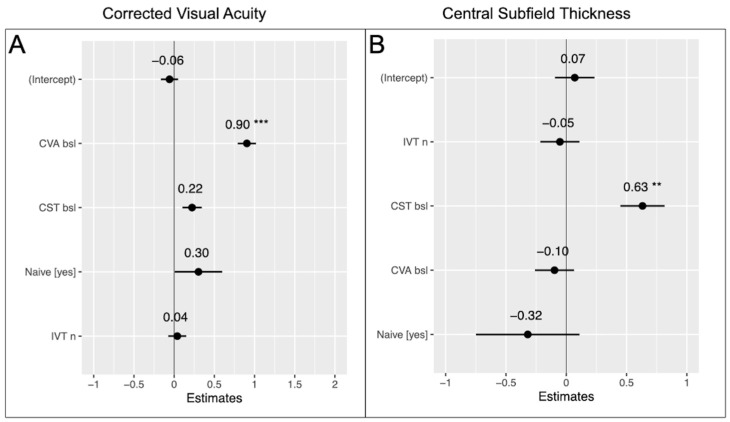
Linear mixed-effects model for (**A**) CVA and (**B**) CST with standardized beta coefficients and 95% confidence intervals of all tested variables. **—*p* value < 0.01, ***—*p* value < 0.001, CVA—corrected visual acuity, CST—central subfield thickness, ETDRS—early treatment diabetic retinopathy study, IVT—intravitreal therapy.

**Figure 3 bioengineering-11-00964-f003:**
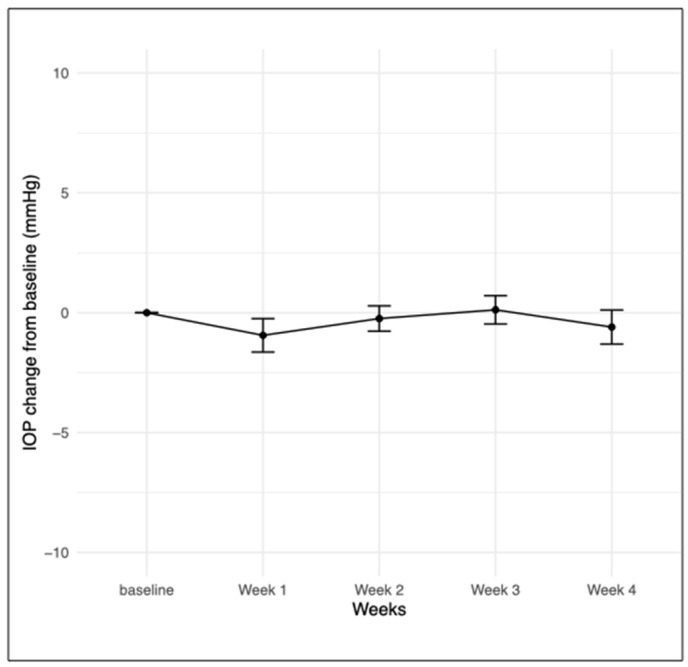
Change in IOP from baseline for eyes with weekly follow-up (n = 23). IOP—intraocular pressure (baseline, week 1 and week 4: n = 23, week 2 and week 3: n = 20).

**Table 1 bioengineering-11-00964-t001:** Baseline characteristics.

Eyes, n	43
Patients, n	31
Women, n (%)	12 (38.0)
Baseline age, years, median [IQR]	66 [58.5–71.5]
Baseline CVA, ETDRS letters, median [IQR]	72 [68.5–78.5]
Baseline CST, micrometers, median [IQR]	325.0 [293.5–399.0]
Baseline IOP, mmHg, mean (±SD)	15.5 (±3.1)
Diagnosis, n (%)	
non-proliferative DR	24 (55.8)
proliferative DR	19 (44.2)
Treatment-naïve eyes (%)	8 (18.6)
Pretreated eyes (%)	35 (81.4)
Median treatment interval, weeks, [min–max range]	4 [4–6]
Previous number of IVTs, median [IQR]	16 [6–28]
Previous anti-VEGF agent, eyes (%)	
Aflibercept	27 (77.1)
Ranibizumab	4 (11.4)
Aflibercept and ranibizumab	4 (11.4)

Table legend: CVA—corrected visual acuity, CST central subfield thickness, DR diabetic retinopathy, ETDRS—Early Treatment of Diabetic Retinopathy Study, IOP—intraocular pressure, IQR—interquartile range, max—maximum, min—minimum, IVT intravitreal therapy, SD—standard deviation, VEGF—vascular endothelial growth factor.

## Data Availability

The data that support the findings of this study are available from the corresponding author, K.F., upon reasonable request.

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
