# Peer review of "Dynamics of Treatment Response to Faricimab for Diabetic Macular Edema"

_bioengineering, 2024, doi:10.3390/bioengineering11100964_

Round 1

Reviewer 1 Report

Comments and Suggestions for Authors

The manuscript add new data of efficacy and tolerance of Faricimab in a weekly basis, in refractory DME patients. Weekly assessment shows that IVT Faricimab is safe. Minor anterior chamber inflammation was found  at W1 and W3. Although the CST decreased, visual acuity remains stable, compare to baseline 

The manuscript is concis , clear and well-writen. 

Reviewer 2 Report

Comments and Suggestions for Authors

Since the weekly follow-up is only available for 23 patients, these data are not necessarily normally distributed.

It might be feasible and of interest to show the exact distribution / change for these patients, e.g. by means of directly connected lines.

Unfortunately, there is no information on excluded patients. PDT is an unusual criterion and brolucizumab was excluded, but other drugs were not. 

What about the other image data (Plexelite)?

While 12 pairs of eyes have been included, there should also be a statistical evaluation just for one eye per patient, besides all eyes treated.

Reviewer 3 Report

Comments and Suggestions for Authors

This manuscript can be significantly improved from the current status. 

The main drawback of the manuscript is that the duration of the study is too short. (This point is captured by the authors themselves under the Discussion in lines 198-203). An optimal duration for this study would be 12-16 weeks, rather than 4 weeks.  

There are several errors in the written language that need to be addressed. These include lines 40, 52-3, 184-7 and 187-191. These are examples only, and not exhaustive.

Comments on the Quality of English Language

The written language can be significantly enhanced.

Reviewer 4 Report

Comments and Suggestions for Authors

The manuscript “Faricimab for diabetic macular edema: weekly efficacy and safety analysis during the first month” by Katrin Fasler and coworkers reports the very early response within 4 weeks after intravitreal faricimab injection in eyes with DME.

-       The first sentence in the abstract is common knowledge and here not relevant, pls delete.

-       The aim of this study to assess “safety of intravitreal therapy (IVT) on a weekly basis for the first month after faricimab” sounds somewhat funny in face of an expected overall risk of mild-to-moderate IOI below 2% over 2 years, a sample size of 32 pts and a 1-month duration of follow up. Though safety reporting is required, I would recommend to delete safety as a study aim and anything referring to safety as a result. No early safety signal reported, that’s it.

-       Details like the address and technical details such as “at the Department of Ophthalmology, University Hospital Zurich, University of Zurich, Switzerland. Approved by the Cantonal Ethics Committee of Zurich, Switzerland.” must not draw the attention from more interesting methodological details and results, for example, given that 77% were pre-treated, the interval since last intravitreal injection. This may explain already in the abstract why no VA gain was observed the CST responded significantly, though less than 10%.

-       The message that “faricimab is an effective treatment for (pretreated) DME showing structural benefit 1 month after the first injection” might become stronger if the few treatment-naïve eyes would be excluded. Early effect defined as no VA gain and a reduction of CST by 8% does in my opinion not justify this statement.

-       Furthermore, anatomic response within the 1st month is nothing unexpected, so that the authors will have to consider, what their findings add to established evidence.

-       It remains to discuss if the sample size justifies the application of linear mixed-effects models to test predictor variables.

-       Numbers in results and abstract differ. IN Table 1, IQR for age displayed as “9” is wrong.

-       Previous number of IVTs, median (IQR) 16 [6 – 28] is wrong given that also treatment-naïve eyes were included.

-       Text in lines 143-4 and 159-60 is not compatible with methods line 74, i.e. weekly assessment was performed in some, but not all patients. Please clarify. Moreover, add the number of observations on the X axis in figs 1A and B.

Taken together, the data inconsistency does not reveal a sufficient quality of this manuscript to support its promotion. It deserves critical work-up before submission including what this study adds to existing knowledge.

Comments on the Quality of English Language

minor english editing required

Round 2

Reviewer 3 Report

Comments and Suggestions for Authors

The authors would need to state the rationale for their study as clearly stated in the rebuttal: i.e. to describe the short-term dynamics of retinal changes as determined by weekly assessments, or something similar.

Comments on the Quality of English Language

The written language has improved significantly in the revised version

Reviewer 4 Report

Comments and Suggestions for Authors

The manuscript received s significant improvement. Beyond the inherent limitations of a small sample size and a short follow up, the authors did well with data analysis  
